# What Are the Roles of Proprotein Convertases in the Immune Escape of Tumors?

**DOI:** 10.3390/biomedicines10123292

**Published:** 2022-12-19

**Authors:** Elham Mehranzadeh, Olatz Crende, Iker Badiola, Patricia Garcia-Gallastegi

**Affiliations:** 1Cell Biology and Histology Department, Faculty of Medicine and Nursery, University of the Basque Country (UPV/EHU), Barrio Sarriena, sn., 48940 Leioa, Spain; 2Nanokide Therapeutics SL, Ed. ZITEK, Barrio Sarriena, sn., 48940 Leioa, Spain; 3Physiology Department, Faculty of Medicine and Nursery, University of the Basque Country (UPV/EHU), Barrio Sarriena, sn., 48940 Leioa, Spain

**Keywords:** proprotein convertases 1, PCs 2, Furin 3, PCSK9 4, tumor immune escape 5, TME 6, cytokines 7, tumor hallmarks 8, immunoediting 9, cancer 10

## Abstract

Protein convertases (PCs) play a significant role in post-translational procedures by transforming inactive precursor proteins into their active forms. The role of PCs is crucial for cellular homeostasis because they are involved in cell signaling. They have also been described in many diseases such as Alzheimer’s and cancer. Cancer cells are secretory cells that send signals to the tumor microenvironment (TME), remodeling the surrounding space for their own benefits. One of the most important components of the TME is the immune system of the tumor. In this review, we describe recent discoveries that link PCs to the immune escape of tumors. Among PCs, many findings have determined the role of Furin (PC3) as a paramount enzyme causing the TME to induce tumor immune evasion. The overexpression of various cytokines and proteins, for instance, IL10 and TGF-B, moves the TME towards the presence of Tregs and, consequently, immune tolerance. Furthermore, Furin is implicated in the regulation of macrophage activity that contributes to the increased impairment of DCs (dendritic cells) and T effector cells. Moreover, Furin interferes in the MHC Class_1 proteolytic cleavage in the trans-Golgi network. In tumors, the T cytotoxic lymphocytes (CTLs) response is impeded by the PD1 receptor (PD1-R) located on CTLs and its ligand, PDL1, located on cancer cells. The inhibition of Furin is a subtle means of enhancing the antitumor response by repressing PD-1 expression in tumors or macrophage cells. The impacts of other PCs in tumor immune escape have not yet been clarified to the extent that Furin has. Accordingly, the influence of other types of PCs in tumor immune escape is a promising topic for further consideration.

## 1. Cancer Hallmarks and PC

The hallmarks of cancer include the complicated progression of tumors that advance to control and stop the body’s normal responses and the existence of human cells that have the capacity to move from a normal condition to a neoplastic state to form malignant tumors [1,2]. The hallmark of cancer contains six distinct biological features identified in the course of the multistep progression of human tumors, and this number has since been expanded to eight [2]. This represents an organizing basis for rationalizing the complexities of neoplastic diseases. These complexities consist of the following: sustaining proliferative signaling, evading growth suppressors, resisting cell death, enabling replicative immortality, inducing angiogenesis, and activating invasion and metastasis. The fundamental reasons for these hallmarks are genome instability, which generates the genetic diversity that expedites their acquisition, inflammation, and fosters multiple hallmark functions. Scientific progress within the last decade has revealed two emerging hallmarks of potential generality that can be added to the present list: the reprogramming of energy metabolism and the evasion of immune destruction [3]. The variety of tumor hallmarks has been defined in different ways by various researchers. Here, we focus on immune evasion, which is known to be an important hallmark in solid tumors [4]. Tumors are surrounded by a tumor microenvironment (TME), which is an extremely complicated ecosystem. Tumor cells interact with immune cells (including macrophages, poly-morphonuclear cells (PMN), mast cells, natural killer (NK) cells, dendritic cells (DCs), and T and B lymphocytes), and non-immune cells (which include endothelial cells and stromal cells) and demonstrate sophisticated interactions with these that regulate the tumor’s natural behaviors. In particular, the immune cells’ components are essential in delineating the tumor’s destiny, their invasive character, and their metastatic capacity. A wide range of different types of immune cells may infiltrate into the tumor, and their formation and organization inside the TME are closely related to the clinical results of patients who struggle with different cancers [5]. The mission of these immune cell types in tumor growth and progression is numerous and is closely related to their inherent functions and to the molecules they express [6]. Moreover, TME includes non-malignant tumor cells such as cancer-associated fibroblasts (CAFs), endothelial cells and pericytes composing tumor angiogenesis, immune and inflammatory cells, bone marrow-derived cells, and the extracellular matrix (ECM) organizing a sophisticated cross-talk with tumor cells [7].

In this review, we try to determine some proteins that have this capacity to change the TME towards immune escape and describe the effects of PCs on these proteins. Immune escape characterized by an incapacitated immune system for the eradication of transformed cells is the hallmark of carcinogenesis [8,9]. There exist proteases such as PCs that interfere in the activation of almost all proteins. Mutations in proteases and/or abnormal protease activity are significantly correlated with several pathological problems such as cancer, Alzheimer’s disease, cardiovascular disorders, and autoimmune diseases [10]. Post-translational changes are paramount strategies that contribute to the biological functions of proteins. One such modification is the endo-proteolysis of precursor proteins, resulting in activation, inactivation, or functional changes [11]. This cleavage procedure can be general or restricted to a few bonds through particular convertases and is followed by amino-terminal, internal, and carboxy-terminal modifications into smaller biologically active polypeptides [12,13]. Proprotein convertases (PCs) are a family of nine serine proteases involved in the processing of cellular pro-proteins. They induce the activation, inactivation, or functional changes in numerous proteins such as neuropeptides, hormones, receptors, and growth factors. Therefore, these enzymes are fundamental for cellular balance in health and disease. Nine PC subtilisin/kexin genes (PCSK1 to PCSK9) encoding for PC1/3, PC2, Furin, PC4, PC5/6, PACE4, PC7, SKI-1/S1P, and PCSK9 are known. The expression of PC1/3, PC2, PC5/6, Furin, and PC7 in lymphoid organs such as lymph nodes, thymus, and spleen has been found to play a role in these enzymes regarding immunity [14]. Proprotein convertases are in the group of proteases that cleave proteins and turn them into their active or inactive form. Several of their substrates are involved in tumorigenesis and immune suppression [15].

## 2. The Role of the Immune System in the TME

The immune system is a complicated set of cells, tissues, and organs. They work together to preserve organisms from dangerous substances, pathogens, and tissue damage and to stop the event of diseases. The immune system has been categorized into innate and adaptive immunity based on variations in the activation of immune responses to many threats [16]. Immune cells are a part of the tumor microenvironment (TME) and the communication between immune cells, other TME cells, and cancer cells plays a principal role in tumor development. Tumors exist in a complex microenvironment in which several kinds of cells can be found [17]. All kinds of immune cells can be found in tumor microenvironments including macrophages; dendritic cells (DCs); mast cells; natural killer (NK) cells; naive and memory lymphocytes B cells; effector T helper (Th) cells including: Th1 cells, Th2 cells, and Th17 cells; regulatory T (Treg) cells; T follicular helper (TFH) cells; and T cytotoxic (TC) cells [18]. These immune cells may be located in the center of the tumor, in the invasive margin, or close to tertiary lymphoid structures (TLSs). Inflammatory and immune cells liberate growth factors (GFs) such as epidermal growth factor (EGF), vascular endothelial growth factor (VEGF-A/C), fibroblast growth factor (FGF2), as well as different cytokines amplifying inflammatory conditions, and enzymes degrading the extracellular matrix, which include MMPs (matrix metalloproteinases), cathepsin, and heparinases [18]. In addition, these cells secrete TGF-B and IL-10, which are associated with immunosuppression by activating and recruiting regulatory T cells (Treg) into the tumor [19,20]. At first, TGF-B1 controls the Th1 and Th2 balance in favor of Th2 phenotypes without cytotoxic activity against the tumor. TGF-B then constrains Th1 response and M1-type macrophage activity; suppresses lymphocytes CD8+, natural killer (NK) cells, and dendritic cell (DC) function; generates Treg with immunosuppressive function; and promotes M2-type macrophages with pro-tumor activity [19,20]. Regulatory T cells are one of the most striking immunosuppressive subsets of CD4+ (CD25+) T cells; they are primarily regulated by master transcription factor 3 (FOXP3), and they account for approximately 5% of the total CD4+ T cell population under conventional circumstances [21]. Tregs, as a prominent mechanism for the controlling equilibrium of the immune system and the immune tolerance of the body, play an essential role in regulating tumor immunity, and they can hinder the activation and differentiation of CD4+ helper T cells and CD8+ cytotoxic T cells to promote reactivity against autologous and tumor-expressed antigens [22,23,24]. In the TME, Tregs can be stimulated and differentiated by conventional T cells, which have a powerful immunosuppressive function, inhibit antitumor immunity, and induce the incidence and development of tumors. Tregs can also suppress the mission of immune effector cells through numerous mechanisms and are key factors in tumor immune escape [25,26,27,28]. The TME provides a suppressive action on Tregs by the overexpression of immune checkpoint (IC) molecules. Targeting IC molecules on Tregs may be effective for cancer treatment. Some of the most important IC molecules are CTLA-4, TIGIT, PD-1, and GITR [29]. Regulatory CD4+ T lymphocytes (Treg) directly secrete or facilitate the formation of immunosuppressive molecules (e.g., IL-10 and adenosine), and modulate the APC function (e.g., via CTLA-4–CD80/86 interactions) [30].

Regarding macrophages in the TME, they are in the group of myeloid cells that have many type of phenotypes in which the M1 or M2 subclasses are the most important. The function of M1 macrophages in immunity against tumors is that M1 cells are “classically activated” by IFNγ, and annihilate tumor cells via their production of nitric oxide (NO) and type 1 cytokines and chemokines. Moreover, M1 acts as an APC, activating cytotoxic CD8+ T cells in an antigen (Ag)-specific manner. Another subcategory is M2 macrophages. M2 cells are activated by “alternative” pathways through IL-4, IL- 13, and/or TGFβ. In tumor immunity M2 has a different action in comparison with the M1 type. M2 releases type 2 chemokines and cytokines; as a result, they elevate the growth of tumors and progression. In the TME, stromal and tumor-associated factors can shift macrophages to the M2 phenotype that are known as polarization, particularly the type of tumor-associated macrophages (TAMs) that promote tumor progression, angiogenesis, and metastasis [31]. Consequently, some immune cells often illustrate flexibility in the TME, demonstrating both tumor-promoting and tumor-hindering potential. For instance, while some macrophages (M1) predominately generate pro-inflammatory cytokines that ensure the anti-tumor immune response, others (M2) can induce the proliferation of fibroblasts, ECM degradation, and immunosuppression [32]. Moreover, tumor-promoting M2 macrophages accompany other cells to enhance tumor progression, such as immature granulocytic and monocytic cells (myeloid-derived suppressor cells (MDSCs)) that can accelerate tumor progression via elevating the proliferation of stromal cells, causing angiogenesis, deposition of extracellular matrix (ECM), and cell migration [33,34,35].

The presence of cytotoxic CD8+ memory T cells in the TME, similar to other immune system cells, destroys tumor cells by identifying neo-antigens (specific antigens on the tumor cells) and provoke an immune response that follows a prototypical, tri-phasic pathway(activation, proliferation, and differentiation of cells). CD8+ T cells in the TME are mainly assisted by CD4+ T helper 1 (TH1) cells that secrete interleukin-2 (IL-2) and interferon-gamma (IFN-γ). Further subclasses of CD4+ cells, for example TH2 cells, collaborate in the B cell response by producing IL-4, IL-5, and IL-13. TH17 cell in the group of the CD4+ subpopulation T cells, on the contrary, generate IL-17A, IL-17F, IL-21, and IL-22, which induce tumor proliferation by favoring tissue inflammation [36]. The influence of the TME on tumor expansion has also been considered according to the impact of B lymphocytes in innate natural killer T (NKT) and natural killer (NK) cells. B lymphocytes normally exist in the draining lymph nodes lymphoid structures neighboring the TME and the invasive tumor margin. The B lymphocytes in the TME play crucial roles in both the control of tumor cell maintenance and the increase in the occurrence of treatment resistance. In brief, these cells have been identified as playing a role in the stimulation of immune escape [36].

Another part of the immune system in the TME is cytokines. Accordingly, it was previously mentioned that these have a close relation with almost all cells as messengers. They are released or membrane-bound proteins that carry out the proliferation, differentiation, and initiation of immune cells. Therefore, uncontrolled cytokine generation is considered to be an important factor in the progression of disorders, for instance, autoimmune diseases and cancer [6,37]. Cytokines can carry out various biological functions of cells, including proliferation, differentiation, and migration. Cytokines that have a weight of up to 70 kDa are identified as small proteins [38]. Based on their construction and mission, they have been categorized into definite super-families, which include interferons (INFs), interleukins (ILs), tumor necrosis factors (TNFs), transforming growth factors (TGFs), chemotactic cytokines (chemokines), and colony-stimulating factors (CSFs) [39]. Within the TME, cytokines can form a tumor-supportive immune microenvironment that suppresses anti-tumor immunity and exerts direct tumor-promoting signals; in contrast, some of them can promote immune escape [40]. In contrast to tumor immune escape (TIE), there is tumor immune surveillance(TIS), in which the immune system recognizes precancerous or cancerous cells and removes them before they can create harm [41]. Additionally, the importance of immune surveillance has been demonstrated in some research. The major molecules involved in tumor immune surveillance, as recognized in knockout mouse models, consist of interferon-g (IFNg); perforin; tumor necrosis factor-related apoptosis-inducing ligand (TRAIL); IL12 and its contributing apoptosis-inducing receptors DR4 and DR5; the recombination activating genes RAG1 and RAG2, which are critical for T cell development; the T cell receptor; and the activating NK cell receptor NKG2D [42]. A deficiency of any of these proteins can lead to more frequent or faster tumor immune escape and more tumorigenesis [43]. Table 1 indicates the effect of some important cytokines in TIS and TIE. We discuss some of these in more detail in this review.

## 3. Tumor Immune Escape as a Part of Immunoediting

The importance of tumor immune escape has been clarified in various hypotheses. In this section we focus on immunoediting as a fascinating area regarding different stages of tumor progression. From approximately the late 19th century, the immune system was considered with increasing interest as a field of study and a novel topic for cancer therapy; a wide range of scientific findings during recent decades in particular determined the essential capacity of the immune system in tumorigenesis. Hypotheses of immunoediting have expressed this ability by delineating three steps of the tumor–immune system interaction: the first step is known as elimination; the second is equilibrium; and the final step is escape, where the progression of tumors moves from activating immunologic surveillance and devastation through active immunologic balance to unrestricted growth. The fundamental purposes of immunotherapy are to limit and decrease progression through these phases, which leads to improvements in the immune system’s responses to modulating tumor growth [46].

### 3.1. Elimination

The first phase of immunoediting is elimination. Immunologic elimination is described in the following several steps: (1) initially, the recruitment of the innate immune system; (2) the presence of cytokines restricting tumor proliferation and vascularization; and (3) direct and indirect anti-tumor immune responses [47]. The goal of the immune system in the elimination phase is to eradicate tumors by utilizing all of its capacity.

### 3.2. Equilibrium

The second phase of immunoediting is equilibrium. Cancer has an ability to grow in spite of the collaboration of the adaptive and innate immune systems, which are initiated in the elimination phase. This equilibrium phase of immunoediting, as characterized by Dunn et al., is a dynamic procedure expressing the immunologic reaction to the primary signs of tumor escape, i.e., a balance between regulation of the immune system and uncontrolled tumor proliferation [48,49].

### 3.3. Escape

The third phase of immunoediting is escape. Due to the continuous flow of immune responses and the production of new indigenous tumor adaptations in the former step (equilibrium), some tumor cells are ultimately selected that have less immunogenicity. Accordingly, tumor cells that emerge in the final phase of immunoediting, i.e., escape, clearly have lower sensitivity to immunologic responses. Consequently, the escape step serves as “rightward push” from the equilibrium phase with the expansion of tumor growth and the decline in immunologic control [50]. There exists a wide range of processes by which tumors can increasingly escape immunologic surveillance [51]. One of the eminent mechanisms that has been at the center of attention during recent decades is the existence of an inhibitory regulatory CD4+FoxP3+ T cell population in higher proportions that predisposes patients to various cancers, probably because regulatory T cells (Tregs) generate cytokines with immunosuppressive traits, for instance, TGF-β and IL-10 that restrict CD8+ T cell activity and differentiation [52,53,54]. The presence of tumor-associated macrophages 2 (TAM2) with CD4+ T lymphocytes releasing IL4 [55] and myeloid-derived suppressor cells (MDSCs) [56,57] moves the TME in favor of tumor immune scape. Furthermore, tumors have been identified as directly inhibiting the cytotoxicity of CD8+ T cells by the upregulation of PI-9, which results in the inactivation of granzyme B (GrB). GrB is a serine protease usually found in the granules of cytotoxic T cells and NK cells that kills tumors and virus-infected target cells [58]. Another important immune escape mechanism that has been under consideration in recent years is when tumors downregulate or totally remove the expression, processing, and presentation procedures of MHC class I; thus, this may contribute to confining the immune system’s capacity to target tumor cells specifically [59,60]. Nevertheless, the loss of MHC class I molecules can predispose tumors to eradication by NK cells. To solve this problem, tumor cells have been shown to downregulate co-stimulatory proteins such as MICA, MICB, and NKG2D (which are receptors for NK cells on tumor cells), supporting a subtle means of escape from NK cells barriers [61,62]. The functions of the innate immune system in suppressing tumor proliferation are via all the three stages of immunoediting, and are often controlled through the Fas and TRAIL death receptor pathways that commence with apoptotic caspase cascades [63].

In recent decades, growing breakthroughs in the molecular mechanisms pertaining to triggering the immune system to defend against the demanding conditions that are created by tumor cells have modified the areas of immunotherapy research. It has been proven that any failure in the host’s immune system responses displays one of the essential processes by which tumor cells evade immune surveillance [64]. These mechanisms related to tumor immune escape have been explained, and the pro-tumor and anti-tumor factors are displayed in a summary in Figure 1.

To become more acquainted with some of the mechanisms of immune surveillance and tumor immune escape, we can study the influences of some molecules precisely. Among these proteins, PC enzymes are principally regulators in host defense. Biochemical analyses have determined that the PCs modulate proteolytic cascades that are crucial in the functional maturation of various proteins critical for the host immune system to defend against challenges, for example, matrix metalloproteinases (MMPs), integrin, and cytokines [65,66]. In the following, the roles of several types of PCs in tumor immune escape are discussed in detail.

## 4. Furin and Tumor Immune Escape

Furin is a PCSK enzyme; it is termed Furin because it is in the upstream region of an oncogene known as FES (the FES upstream region). Investigations have shown the importance of this enzyme in a wide range of biological procedures. Furin has been considered a master switch of tumor progression and proliferation [67,68].

Enhancing the activation of Furin can support cancer expansion by suppressing preservative anti-tumor mechanisms. For instance, extending the activation of Furin-mediated TGF-B lowers immune surveillance by stimulating the expansion of some suppressive cells such as Treg cells and by inhibiting effector T-cell functions [69]. TGF-B1, as an anti-inflammatory cytokine, also mediates in the support of peripheral tolerance and preservation in autoimmune circumstances. Normally, TGF-B1 is synthesized as an inactive form that is pro-TGF-B1 and should be degraded to produce the mature and biologically active cytokine. Much research has proven that Furin is vital for the maturation of pro-TGF-B1 in Tregs [70,71]. Moreover, Ag presenting cells (APCs) are impeded by the activity of Tregs; thus, Tregs repress the immune system via cytotoxicity and disruption of metabolic-related pathways [72].

In several studies, knockout of the Furin gene in T cells was found to be related to the release of a high number of pro-inflammatory cytokines (IL6, TNFα, and IL1 β) and autoantibody products in mice. Therefore, this shows the pivotal role of this protein in immune tolerance. Furthermore, Furin, via its proteolytic activity, manages the suppressive functions of Tregs and thus prevents chronic inflammation and autoimmune diseases. In other cells, such as macrophages, Furin mediates in the modulation of their inflammatory phenotype (M1) and has an important role in the defense against tumors [14]. Subsequently, Furin mediates in the activation control of macrophages. Indeed, deprivation of Furin in macrophages indicates positive regulation of many genes implicated in their activation such as Serpinb1a, Serpinb2, Ccl2, Ccl7, Il6, and Il1-b [73]. Deleting the Furin gene combined with LPS or IFN-γ stimulation is linked with over expression of Nos2, a pro-inflammatory phenotype marker of macrophages (M1). Contrary to this, the expression of the anti-inflammatory and Arg1 phenotype marker was found to be decreased [73]. Thus, Furin plays an anti-inflammatory role in these cells and changes the macrophage phenotypes in TME from M1 towards M2. The mission of macrophage M2 cells is to influence immune escape activity by reducing DC maturation and T cell effector function [74]. On the other hand, M1 macrophages increase the activation of T cells, suppress the recruitment of M2 cells, and regulate the normalization of tumor angiogenesis [75].

In one study, T cells that underwent the deletion of Furin did not promote any major defects, and, consequently, although the mouse models were alive, these mice displayed signs of inflammation and fibrosis. In T cells, the deletion of Furin did not interfere in their development but did noticeably decrease their capacity to release anti-inflammatory cytokines, IL10, and transforming growth factor beta 1 (TGF-B1) [70].

Transforming growth factor (TGF-B) is a fundamental regulator of immune tolerance and balance, hindering the development and operation of many substrates of the immune system. The signaling dysfunction in TGF-B is associated with the promotion of the emergence of tumors and inflammatory diseases. Additionally, TGF-B plays an important role in immune suppression inside the tumor microenvironment, and, in recent years, research has revealed its function in tumor immune escape and imperfect responses to cancer immunotherapy [76]. The knockout of Furin also contributes to increased production of IL-6 and IFN-g as pro-inflammatory cytokines. On the whole, positive regulation of some genes such as FOS, JUN, and IFN-G may lead to T cell activation [70].

According to these studies, it can be inferred that PCs interfere in tumor immune evasion by activating or inactivating cytokines. These cytokines are released or bound on the cell membrane and adjust the growth, differentiation, and activation of immune cells. Thus, any perturbations in cytokine production can be linked to various diseases such as cancer and autoimmune disorders [6,77].

## 5. Furin and MHC Class 1 Regulation

The complex of major histocompatibility class I (MHC I) and peptides derived from a cell’s expressed genes that convey and present this antigenic information on the surface of all nucleated cells is an important part of the immune system. This complex enables CD8+ T lymphocytes to recognize pathological cells that generate aberrant proteins, for instance, tumors that represent mutated peptides. In this condition, for tumors to retain and continue their proliferation and progression, it is necessary to utilize mechanisms to avoid elimination by CD8+ T lymphocytes. This is due to the fact that MHC I molecules are not vital for cell survival; accordingly, one procedure by which tumors may escape immune control is by eliminating the MHC I antigen presentation machinery (APM). Consequently, this has two aspects: the first is to diminish the capability of natural immune responses to control cancers, and the second is to inhibit immune therapies that work by re-stimulating anti-tumor CD8+ T cytotoxic, such as checkpoint blockades [78]. MHC-I ligands are chiefly generated by proteasomes; however, some MHC-I ligands are also produced in the trans-Golgi network. Strikingly, it has been confirmed that, in this part, Furin interferes in the proteolytic cleavage and the release of antigenic peptides [79,80]. Furin, a proprotein convertase, is located mostly in the trans-Golgi network; it intervenes in the maturation of numerous proproteins by cutting them at precisely three to four basic residues [81]; however, its exact role in increasing or decreasing the expression of MHC1 in normal, tumor, and immune system cells such as lymphocytes and APCS is under investigation.

## 6. Furin and Checkpoints

In tumors, the response of T cells is interrupted through the identification of PD1 (programmed cell death protein 1), also known as the CD279 (cluster differentiation) receptor, exhibited on the surface of T lymphocytes and B cells that has some roles in immune system responses, mainly promoting suppression. The expression of its ligand, termed PD-L1, is high on the surface of tumor cells and pro-tumor macrophages [82]. PD-1 is an important marker on exhausted T lymphocytes and acts as a co-inhibitory checkpoint [83]. In an exhausted condition, T lymphocytes are terminally differentiated and express a high and constant number of inhibitory receptors, such as PD-1 [84,85]. The presence of PD-1 is enhanced exponentially with tumor proliferation exponentially [83].

Inhibiting the expression of Furin is a useful means of accelerating an anti-tumor response. Eliminating PCs can suppress PD-1 expression, thereby obstructing the cleavage and operation of the Notch–Delta signaling pathway and boosting the activation and growth of T lymphocytes, their viability, and their cytotoxicity against both MSI (microsatellite instability) and MSS (microsatellite stable) colon tumor cells. Likewise, the knockout of PCs may enhance tumor-infiltrated CTLs, which increases immune response in the TME against tumor cells and can also boost tumor regression [86]. After activating T cells, some transcription factors, such as the nuclear factor of activated T cells (NFAT), T-bet, Blimp-1, and FoxO1, induce and regulate the expression of PD1 [87]. Diverse intracellular signal transduction pathways control the activation and expression of these factors. They consist of anti-inflammatory signals, tyrosine kinases, and mitochondrial apoptosis pathways. Numerous proteins are recruited in these intracellular signal pathways that need to undergo proteolytic cleavage of their precursor forms through the proprotein convertases (PCs) to make them biologically active [88].

The elimination of PCs suppresses PD-1 expression via obstructing proteolytic maturation of the Notch precursor, blocking calcium/NFAT and NF-kB signaling, and increasing the activation of the ERK protein. This information demonstrates the principal role of PCs in controlling the expression of PD-1 and supports the idea of targeting PCs as a contributing approach to tumor immunotherapy [86].

## 7. The Role of PC1/3 and PC7 in Immune Escape

Many findings have indicated the role of Furin as one of the protein convertases in immune modulation, but there are some correlations with other PCs. According to de Zoeten’s studies, it was noted that one of the key transcription factors in the regulation of Treg functions is Foxp3 [89]. There are some potential cleavage sites for PCs that are found on the protein sequence of Foxp3. In mice Tregs, these cleavage sites were detected on Foxp3 transcription factors that can be proteolytically activated by PC1/3 and PC7. In addition, the secretion of IL-10 was found to be increased in Tregs over expressing the truncated Foxp3 form. These findings highlight that PC1/3 and PC7 regulate the immunosuppressive functions of Tregs by Foxp3 cleavage [89].

## 8. PCSK9 and Immune Escape

The role of PCSK9 is mainly to control the levels of cholesterol in the body, and it is associated with its capability to downregulate LDLRs (low-density lipoprotein receptors) on the cell surface by conducting LDLRs for degradation to the lysosome rather than redirecting them back to the cell surface via both extracellular and intracellular pathways [90,91,92,93,94]. Therefore, PCSK9 leads to the lowering of the cholesterol metabolism [95]. In accordance with some findings, it has been clarified that inhibiting PCSK9 may induce the infiltration of T lymphocytes in to the TME, subsequently provoking tumor cells to respond to immune checkpoint therapy [95]. Moreover, studies have determined that the number of active T lymphocytes positively contributes to the success of immune checkpoint blockade therapy. In one study, the flow cytometry method was applied, and the data indicated a noticeable boost in numbers of infiltrating CD8+ cytotoxic T cells (CTLs), natural killer (NK) cells, CD4+ T helper (TH) cells, and γδ T cells in tumors with a deficiency in PCSK9 genes. Contrarily, a remarkable increase in the number of CD4+ Foxp3+ regulatory T (Treg) cells was not observed. Additionally, there were no amplifications in the number of CD8+or CD4+ T lymphocytes in the host mice’s spleens. The proportion of CTLs to Treg cells notably increased in the tumors that underwent the elimination of PCSK9. Another finding is that the number or proportion and percentage of interferon-γ+ (IFN-γ+) and granzyme B+ (GZMB+) CTLs was also strikingly enhanced in PCSK9-absent tumors [95]. These findings regarding the presence of INFG and GZMB in tumors without PCSK9 indicate the paramount role of PCSK9 in immune suppression in tumors.

## 9. PCSK9 and Expression of MHC1 on the Surface of Tumors

MHC class 1 is expressed on almost all nucleated cells [96]. In one study, it was found that PCSK9 manages the expression of MHC I on the surface of nucleated cells. The overexpression of PCSK9 in cells leads to more H2-K1 (major histocompatibility class 1 in a murine model) being localized in the lysosome, instead of being located on the plasma membrane and then degraded. On the contrary, with the knockout of PCSK9 in mouse cells, staining methods showed increased localization of H2-K1 in the plasma membrane [95]. As a result, it can be inferred that PCSK9 has a crucial role in regulating MHC class 1 on the cell surface and can thus influence immune infiltration in the TME [95]. To broaden our understanding pertaining to PCSK9 as well as other PCs and the expression of MHC class 1 on tumor cells, more research should be carried out to find key factors in presenting antigens that are more influenced by PCs in tumor immune escape responses.

## 10. Conclusions

In recent years, the use of immunotherapy and its links with immunoediting in the treatment of different kinds of cancers has been applied [97,98,99]; however, some patients have shown resistance to this type of treatment and scientists have tried to find subtle ways to alleviate the problems related to immunotherapy resistance and boost its efficiency. For instance, individual tumor studies for the specific treatment of tumors are under exploration; thus, to achieve this purpose, tumor biology needs to be investigated to allow the development of accurate molecular treatments. We reviewed data regarding different protein convertases interfering in the immune aspects of the TME, and our findings indicate the paramount role of these proteins. Recognizing the proteins that are relevant to the immunoediting phases and that can push tumors from the immune escape phase to the equilibrium or elimination phases would be a promising way to augment immunotherapy efficacy since one purpose of immunotherapy is to maintain tumors in the equilibrium or elimination phases and prevent further progression to the escape phase. We recommend that more research should be carried out to further clarify the role of PCs in tumor immune escape.

## Figures and Tables

**Figure 1 biomedicines-10-03292-f001:**
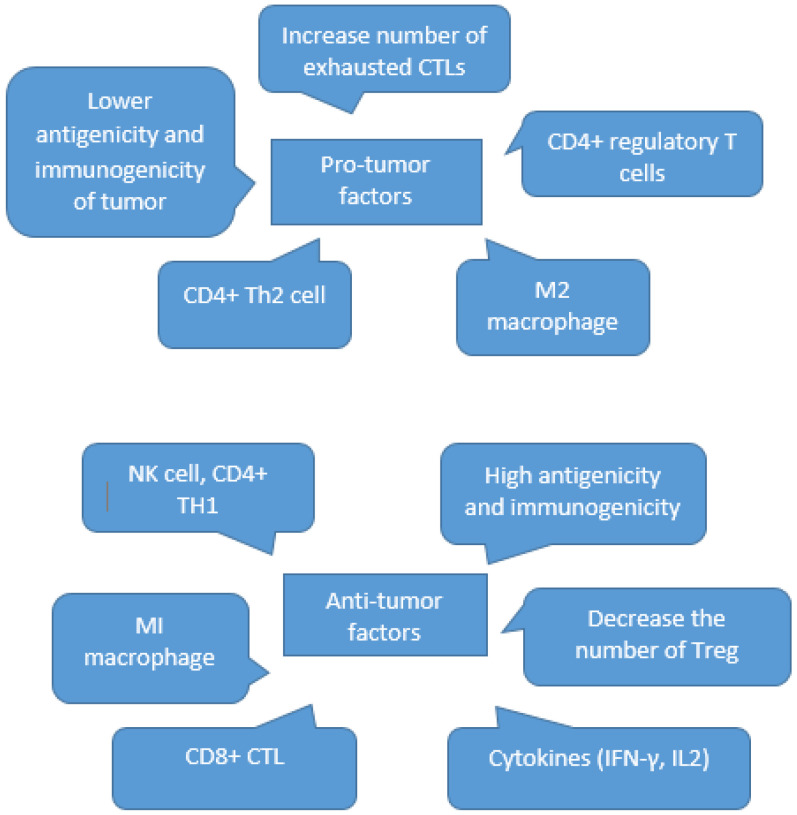
Summary of pro- and anti-tumor factors [31].

**Table 1 biomedicines-10-03292-t001:** The effects of some cytokines in tumor immune surveillance and tumor immune escape [44,45]. TIS: tumor immuno-surveillance, TIE: tumor immuno-escape.

Cytokines	Effect	Released by
IL-10	This is related to immune suppression, tumors expansion, and hindering of Th1 response by inhibiting the release of pro-inflammatory cytokines. In addition, it augments the migration capacity of cells and decreases the expression of MHC class I (TIE).	TH cells, B lymphocytes, macrophages and activated monocytes, thymocytes, and tumor cells.
IFN-γ	Creates a response against tumors by regulating the innate and adaptive immune system. Naturally preserves the organs from the sudden development of tumors (TIS).	B cells, T cells, NK and natural killer T (NKT) cells, mast cells, and macrophages.
IL-12	It has been investigated as an anti-tumor cytokine; it interferes in the migration and metastasis of tumor cells; it is executed through IFN-γ with anti-angiogenic potential; it increases the expression MHC class I and II and increases NK and T CD8+ cell activation. (TIS)	Antigen presenting cells (APCs): macrophages, monocytes, and dendritic cells (DCs).
GM-CSF	It has anti-tumor features by increasing immune system responses and stimulating specific cell-mediated cytotoxicity against autologous tumor targets (TIS).	T cells, monocytes, and macrophages.
TNF-α	Related to tissue devastation, leading to tumor progression and damage recovery, which has anti-tumor activity. May stop DNA repair, acts as a growth factor (GF) for tumor cells, and can induce angiogenesis. Destroy tumor vasculature and has necrotic impacts on tumors at high doses (TIS/TIE).	Tumor cells, T cells, and activated macrophages.
IL-6	It can lead to anti-tumor responses and to tumor progression. It depends on the pathways of the cells that would be activated; it can promote tumor growth, metastasis, and resistance to chemotherapy in different tumor cells (TIS/TIE).	Macrophages, B and T lymphocytes, and keratinocytes.
IL-27	Contributes to apoptosis of cancer cells, increases cytotoxic T cell and NK cell responses, antigen presentation, and Th1 response; decreases proliferation, migration, and invasion of cancer cells; influences Treg development; increases PD-L1 expression levels and cancer cell proliferation (TIS/TIE).	Antigen presenting cells (APCs).
IL-30	Participates in decreasing Th1 cell differentiation, increases cancer cell proliferation, and manipulates IL-27 signaling (TIS).	Activated DCs, NK cell, and T cell.
IL-35	It can interfere in increasing angiogenesis and metastasis, immune suppression, and T cell exhaustion and the proliferation of cancer cells; it decreases Th1 cell differentiation and CTL responses. It can also participate in decreasing migration and invasion of cancer (TIS/TIE).	Secreted primarily by Treg cells, B cells, endothelial cells, smooth muscle cells, DCs, and monocytes.

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
