# Peer review of "What Are the Roles of Proprotein Convertases in the Immune Escape of Tumors?"

_biomedicines, 2022, doi:10.3390/biomedicines10123292_

Round 1
Reviewer 1 Report
The review article entitled “What are the roles of proprotein convertases in immune escape 2 of tumors?” describes the function of protein convertases in tumor microenvironment. The authors have made great efforts to provide a understanding of different protein convertases, their function in tumor. However, the manuscript requires a lot of major revisions before it could be considered for publication in the Biomedicines journal. Please find my comments below to improve the manuscript.
Major-
1. Throughout the manuscript there are numerous typos, broken phrases and inaccurate sentences. Please revise the manuscript throughout carefully for improving grammar and preventing typos to ensure correct meaning of each sentence is displayed.
2. Abstract contains a lot of sentences and is not concise and it should be revised to reflect the purpose and content of the review article accurately.
2. A graphical diagram showing the function of protein convertases in tumor microenvironment would be more beneficial to readers as compared to table.
3. Table 2 could be replaced by a schematic to make it look more appealing.
4. English should be revised throughout the manuscript.
Please review the suggested changes below.
Line 17: Please replace “Last discoveries” to recent discoveries.
Line 19: Please correct TGF-B.
Line 21: Do you mean furins contribute to impairment of DCs and T effector cells? If yes, the sentence needs to be revised accurately.
Line 54-55: Please check this sentence again, it sounds incorrect. Also please minimize the use of a lot of plural words, for eg. Immune cells, constituents, tumors, etc.
Line 65 and 231: Please replace “scape” to escape.
Line 68: Please check this sentence, and capitalize the first letter “Mutations”
Line 69-70: Please check the sentence, it sounds incorrect. You may want to replace “consist of” to “such as”.
Line 105-109: Please check this sentence again. It sounds incorrect.
Line 119: Please correct the spelling “tumours”.
Line 122-123: Please revise it.
Line 154: Do you mean “TH17 cells and other CD4+ subpopulation”?
Line 83-84 and 169: Please check the sentence and
Line 335: Space is missing. Please look for other places throughout the article, there were few more places.
Author Response
Dear madam or sir,
Firstly, we are very thankful for the valuable correction and your guidance. We tried to correct all mistakes that you have highlighted. You could find your improvements to the manuscript in yellow, and the checked list attached in the file.

Reviewer 2 Report
The manuscript authored by Mehranzadehb et al. addresses a very interesting topic regarding the roles of proprotein convertases in immune escape of tumors. Although the topic is very interesting a lot of clarifications still need to be addressed before the manuscript is considered for publication:
1. Written English represents a major problem, the manuscript has some phrases with no coherence: for example the first sentence of the introduction, the first sentence of chapter 2 and many other throughout the text. Please make sure a native English speaker is reading the manuscript before submission.
2.The manuscript reads like an events list rather than a summary of the knowledge in the field presenting the vision of the authors on this matter.
3. The link between TME description and PCs and the transition between chapters is not well described.
4. The first three chapters should be condensed since they do not represent the declared focus of the manuscript.
5. Again, there are meaningless phrases in the entire manuscript, for example:
„In another cells like in macrophages, Furin is mediated in the modulation of their inflammatory phenotype„
„Complex of major histocompatibility class I (MHC I) proteins and peptides which derived from a cell’s expressed genes and therefore convey and present this antigenic in formation on the surface of all nucleated cells„
„PCSK9 manage the expression of MHC I on the surface of nucleated cells, the overexpression of PCSK9 in cells, more H2-K1 (major histocompatibility class 1 in murine model) was localized in the lysosome and then degraded and less in the plasma membrane”
6. These are just a few examples, the manuscript needs serious revision before publication.
Author Response
Dear madam or sir,
Firstly, many thanks for the time you have dedicated to our manuscript, we are very grateful for your valuable feedback. You declared important points related to some sentences seem meaningless. We revised the whole manuscript conscientiously and correct the mistakes. Some of your corrections are highlighted in grey. Moreover, the paper has passed a native English correction service, from MDPI, which has thoroughly improve the paper.
The checked list of improvements is attached in the file.
Yours sincerely

Round 2
Reviewer 1 Report
Thank you for making the necessary revisions. The revised review article looks good.
Reviewer 2 Report
Elham Mehranzadeh and colleagues have improved a lot the quality of the manuscript: "What are the roles of proprotein convertases in tumor immune escape?", but some minor spelling changes still need to be made.
Ex::
Line:71: „In this review, we have tried to determine some proteins”
„In this review, we have tried to address some proteins”
Line358: „tumors that represent mutated peptides”, need to be changed as:
„tumors that present mutated peptides”
Please keep the consistency of pro-protein or proprotein throughout entire manuscript.
In this format I suggest publication of the manuscript.